# A Machine Learning Approach to Determine Risk Factors for Respiratory Bacterial/Fungal Coinfection in Critically Ill Patients with Influenza and SARS-CoV-2 Infection: A Spanish Perspective

**DOI:** 10.3390/antibiotics13100968

**Published:** 2024-10-14

**Authors:** Alejandro Rodríguez, Josep Gómez, Ignacio Martín-Loeches, Laura Claverias, Emili Díaz, Rafael Zaragoza, Marcio Borges-Sa, Frederic Gómez-Bertomeu, Álvaro Franquet, Sandra Trefler, Carlos González Garzón, Lissett Cortés, Florencia Alés, Susana Sancho, Jordi Solé-Violán, Ángel Estella, Julen Berrueta, Alejandro García-Martínez, Borja Suberviola, Juan J. Guardiola, María Bodí

**Affiliations:** 1Critical Care Department, Hospital Universitari Joan XXIII, 43005 Tarragona, Spain; lauraclaverias@gmail.com (L.C.); sitrefler@yahoo.es (S.T.); julen.berrueta@estudiants.urv.cat (J.B.); alejgarcia.hj23.ics@gencat.cat (A.G.-M.); mbodi.hj23.ics@gencat.cat (M.B.); 2Faculty of Medicine, Universitat Rovira & Virgili, 43005 Tarragona, Spain; josep.goal@gmail.com (J.G.); ffgomez.hj23.ics@gencat.cat (F.G.-B.); afranquet.hj23.ics@gencat.cat (Á.F.); 3Pere Virgili Health Research Institute, 43005 Tarragona, Spain; 4Centre for Biomedical Research Network Respiratory Diseases (CIBERES), 43005 Tarragona, Spain; 5Technical Secretary’s Department, Hospital Universitari Joan XXIII, 43005 Tarragona, Spain; 6Department of Intensive Care Medicine, Multidisciplinary Intensive Care Research Organization (MICRO), St James’ Hospita, D08 NHY1 Dublin, Ireland; drmartinloeches@gmail.com; 7Critical Care Department, Hospital Universitari Parc Tauli, 08208 Sabadell, Spain; emilio.diaz.santos@gmail.com; 8Medicine Faculty, Universitat Autónoma de Barcelona, 08193 Sabadell, Spain; 9Critical Care Department, Hospital Dr. Peset, 46017 Valencia, Spain; zaragoza_raf@gva.es; 10Critical Care Department, Hospital Son Llatzer, 07198 Palma de Mallorca, Spain; marcio.borges.sa1967@gmail.com; 11Microbiology/Clinical Analysis Laboratory, Hospital Universitari de Tarragona Joan XXIII, 43005 Tarragona, Spain; 12Center for Biomedical Research in Infectious Diseases Network (CIBERINFEC), 28220 Madrid, Spain; 13Postgrado Medicina Crítica y Cuidado Intensivo, Facultad de Medicina, Fundación Universitari Ciencias de la Salud, Distrito Especial, Cra. 54 No.67A-80, Bogotá 111221, Colombia; carlosmariogonzalezgarzon@hotmail.com (C.G.G.); llcortes@fucsalud.edu.co (L.C.); 14Internal Medicine Department, Hospital Dr. Alejandro Gutiérrez, Venado Tuerto S2600, Argentina; mafloales@gmail.com; 15Critical Care Department, Hospital Universitrio y Politécnico La Fe, 46026 Valencia, Spain; sancho_sus@gva.es; 16Critical Care Department, Hospital Dr. Negrin, 35010 Las Palmas de Gran Canaria, Spain; jsolvio@gmail.com; 17Critical Care Department, University Hospital of Jerez, INIBiCA, 11407 Jerez, Spain; litoestella@hotmail.com; 18Faculty of Medicine, University of Cádiz, 11407 Jerez, Spain; 19Tarragona Health Data Research Working Group (THeDaR), 43005 Tarragona, Spain; 20Critical Care Department, Hospital Universitario Marqués de Valdecilla, 39008 Santander, Spain; borjasuberviola1977@gmail.com; 21Robley Rex VA Medical Center, University of Louisville, Louisville, KY 40202, USA; juan.guardiola@va.gov

**Keywords:** influenza A (H1N1), COVID-19, bacterial coinfection, fungal coinfection, machine learning

## Abstract

**Background**: Bacterial/fungal coinfections (COIs) are associated with antibiotic overuse, poor outcomes such as prolonged ICU stay, and increased mortality. Our aim was to develop machine learning-based predictive models to identify respiratory bacterial or fungal coinfections upon ICU admission. **Methods**: We conducted a secondary analysis of two prospective multicenter cohort studies with confirmed influenza A (H1N1)pdm09 and COVID-19. Multiple logistic regression (MLR) and random forest (RF) were used to identify factors associated with BFC in the overall population and in each subgroup (influenza and COVID-19). The performance of these models was assessed by the area under the ROC curve (AUC) and out-of-bag (OOB) methods for MLR and RF, respectively. **Results**: Of the 8902 patients, 41.6% had influenza and 58.4% had SARS-CoV-2 infection. The median age was 60 years, 66% were male, and the crude ICU mortality was 25%. BFC was observed in 14.2% of patients. Overall, the predictive models showed modest performances, with an AUC of 0.68 (MLR) and OOB 36.9% (RF). Specific models did not show improved performance. However, age, procalcitonin, CRP, APACHE II, SOFA, and shock were factors associated with BFC in most models. **Conclusions**: Machine learning models do not adequately predict the presence of co-infection in critically ill patients with pandemic virus infection. However, the presence of factors such as advanced age, elevated procalcitonin or CPR, and high severity of illness should alert clinicians to the need to rule out this complication on admission to the ICU.

## 1. Introduction

During the last 20 years, two pandemics (influenza A (H1N1)pdm09 and SARS-CoV-2) have strained healthcare systems and caused a significant number of deaths worldwide [1,2,3,4]. It has been observed that, during the last pandemic, a high number of patients (between 60 and 90%) received antibiotics/antifungals empirically to treat a possible or probable bacterial/fungal coinfection (COI) on admission to the ICU [5,6,7,8]. This indiscriminate and inappropriate use of antibiotics led to increased resistance, especially to Gram-negative bacilli [9,10,11].

Several studies [5,7,12,13,14] have reported a variable incidence of respiratory bacterial/fungal COI in patients with influenza A (15–25%) and coronavirus disease 2019 (COVID-19) (8–15%). However, most authors agree that bacterial and especially fungal COI is associated with an increase in the patient’s length of stay and days of invasive mechanical ventilation (MV), and especially with higher mortality in the ICU [5,6,7,8,9,10,11,12,13,14].

Multiple studies [5,6,7,8,9,10,11,12,13,14] have observed different factors associated with COI. Other biomarkers, especially procalcitonin (PCT), have proven useful in predicting or ruling out COI in patients with influenza A (H1N1)pdm09 [6,15,16] or COVID-19 [15,17]. However, these studies have yet to be internally or externally validated and there are many differences among population types that make it difficult to generalize conclusions.

Although these pandemics are a thing of the past, the circulation of pandemic viruses, with their current variants, continues over time and conditions the admission of a non-negligible number of patients to hospitals and ICUs [18,19,20], especially those with impaired immunity [21,22,23].

Having factors that allow us to predict the presence of COI on admission to the ICU could be very helpful in avoiding the inappropriate use of antimicrobials/antifungals, decreasing harmful side effects, and perhaps improving the outcomes of critically ill patients.

The development of predictive models using new machine learning techniques may allow us to determine which risk factors may be associated with COI in critically ill patients. We hypothesize that there are risk factors at ICU admission that can predict the presence of COI. Therefore, the aim of this study is to develop different prediction models using machine learning to predict the presence of COI at ICU admission in an extensive national database.

## 2. Materials and Methods

### 2.1. Design

This is a secondary analysis of 2 prospective, multicenter, observational cohort studies. The first database is the GETGAG registry (Spanish Severe Influenza A Working Group), a voluntary registry created by the Spanish Society of Intensive Care Medicine (SEMICYUC) in 2009 during the A (H1N1)pdm09 influenza pandemic in which 184 Spanish ICUs participated between June 2009, and June 2019 [6,16].

The second database is the COVID-19 registry, a voluntary registry created by SEMICYUC in 2020 during the SARS-CoV-2 pandemic in which 74 Spanish ICUs participated between 1 July 2020, and 31 December 2021 [7,17,24,25].

We reported results in accordance with the Strengthening the Reporting of Observational Studies in Epidemiology (STROBE) guidelines (Appendix A).

### 2.2. Study Population

A total of 8902 consecutive patients requiring ICU admission with a diagnosis of respiratory infection by influenza A (H1N1)pdm09, seasonal A or B (n = 3702), or SARS-CoV-2 (n = 5200) viruses were included during the two periods described.

The presence of the virus was determined by performing reverse transcription-polymerase chain reaction (RT-PCR) in each hospital, according to the Infectious Diseases Society of America (IDSA) recommendations for influenza [26] and World Health Organization (WHO) recommendations for SARS-CoV-2 [27]. The follow-up of patients was scheduled until confirmed ICU discharge or death, whichever occurred first.

### 2.3. Definitions

Coinfection was suspected if a patient presented with signs and symptoms of lower respiratory tract infection, with radiographic evidence of a pulmonary infiltrate that had no other known cause [5,6,7].

Coinfection had to be confirmed by laboratory testing using Centers for Disease Control and Prevention (CDC) criteria [5,6,17]. Only respiratory infection that was microbiologically confirmed with a respiratory specimen or serology obtained within two days of ICU admission was considered community-acquired COI.

Lower respiratory tract infections (LTRIs) diagnosed after two days of ICU admission were considered nosocomial superinfection and excluded from the present study.

The diagnosis of COI was considered “definitive” if respiratory pathogens were isolated from blood or pleural fluid and if serological tests confirmed a fourfold increase in atypical pathogens, including *Chlamydia* spp., *Coxiella burnetiid*, and *Mycoplasma pneumoniae*. The diagnosis of COI was considered “probable” when a respiratory pathogen was isolated from a respiratory specimen (bronchoalveolar lavage (BAL) or tracheal aspirate (TA)) or was positive for *S. pneumoniae* or *Legionella pneumophilia* in urine antigen test. Sputum was not used as a respiratory specimen for the diagnosis of LRTI. Only patients with a definite or probable diagnosis were included in the present analysis [5,6].

The diagnosis of pulmonary aspergillosis (PA) was defined according to the recently modified criteria proposed by Verweij et al. [28]. “Proven” PA was defined by lung biopsy showing invasive fungal elements and growth of *Aspergillus* spp. in culture or positive PCR for *Aspergillus* spp. in tissue. “Probable” PA was defined by pulmonary infiltrate and bronchoalveolar lavage (BAL) culture or cavitating infiltrate and sputum/sputum positive for *Aspergillus* spp. “Possible” PA was defined with lung infiltrate and positive tracheal aspirate or mini-BAL culture [28,29].

Diagnostic procedures for actively searching for *Aspergillus* spp. or other bacteria were not standardized and were requested at the discretion of the treating physicians. Galactomannan levels in BAL or serum are not available in our general registry.

### 2.4. Study Variables

Demographic data, comorbidities, and clinical and laboratory findings were collected during the first 24 h of ICU admission. In addition, the need for invasive mechanical ventilation and the presence of shock on admission to the ICU were recorded.

Disease severity was determined using the Acute Physiology and Chronic Health Evaluation II (APACHE II) score and the level of organ dysfunction was evaluated using the SOFA score. The variables that were controlled for in this study can be seen in Table 1.

### 2.5. Analysis Plan and Statistical Analysis

First, the incidence of COI in the general population was determined and patient characteristics were compared between the COI and non-COI groups. Qualitative variables were expressed as percentages, while quantitative variables were expressed as median and interquartile range Q1–Q3. Chi-square and Fisher tests for categorical variables and Student’s *t*-test or Mann–Whitney U test for quantitative variables were used to determine clinical differences between groups.

Secondly, binary logistic regression was employed to ascertain which variables were independently associated with COI. All variables with statistical significance (*p* < 0.05) in the bivariate comparison between groups were included in the GLM (generalized linear model). Furthermore, the SOFA score was incorporated into the final model, deemed a crucial factor from a clinical standpoint despite failing to achieve statistical significance in the comparative table.

Given the significant discrepancy between the two groups, with only 14% of patients being in the COI group, it is essential to consider this imbalance when developing the model to ensure optimal performance. The ROSE (Random Over-Sampling Examples) package was implemented to address the imbalance between the groups. This statistical package provides functions to address binary classification issues in the context of unbalanced classes. Balanced samples are generated using a smoothed bootstrap approach, allowing for estimation and accuracy evaluation of a binary classifier in the presence of a rare class. The package also includes functions that implement more traditional remedies for class imbalance and various accuracy evaluation metrics. These are estimated using holdout, bootstrap, or cross-validation methods [30,31]. The ‘under’ option simply determines under-sampling without replacing the majority class until either the specified sample size N is reached or the positive examples have a probability p of occurring. When the method is set to ‘under’, the resulting sample will be reduced.

The ROSE software (version 0.0-4) was used exclusively for data processing of the Train subset, with the Test subset remaining untouched. Once the model had been developed in the Train set, it was applied to the Test set and its performance was evaluated. Results are presented as odds ratio (OR) with 95% confidence interval.

The performance of the model was assessed by determining accuracy, precision, sensitivity, specificity, and area under the ROC curve (AUC). The Hosmer–Lemeshow goodness of fit was also determined and the presence of collinearity between the explanatory variables was examined using variance inflation factors (VIFs).

In addition, we performed k-fold cross-validation (K = 10), which consists of taking the original data and creating two separate sets from it: a primary development set (Train), and a secondary validation set (test). The training set is then divided into k subsets, and, at the time of training, each k subset will be taken as the model test set, while the rest of the data will be taken as the training set. Once the iterations were completed, the accuracy and error were calculated for each of the models produced, and to obtain the final accuracy and error, the average of the k-trained models was calculated.

Finally, we studied the normality of the residuals of the model. A residual measures the vertical distance between a point and the regression line. Simply put, it is the error between a predicted value and the actual observed value. The most important assumption of a linear regression model is that the errors are independent and normally distributed. The normality of the residuals was assessed visually using different plots and by applying the RESET (Regression Equation Specification Error Test), which assesses the adequacy of a linear regression model by including polynomial terms of the independent variables. A significant *p*-value rejects the linearity hypothesis, as a result of which it is concluded that it is not fulfilled, meaning that a non-linear regression model should be chosen [32].

Third, we developed a random forest classifier model (RFc). Random forest models are a powerful tree-based non-linear learning technique in machine learning. The model developed was set to perform 500 random trees, with a minimum number of 15 variables per tree.

The performance of the RFc model was evaluated using the out-of-bag (OOB) error. This method allows the prediction error of random forests, boosted decision trees, and other machine-learning models to be measured using bootstrap aggregation. We also plotted the importance of different variables to the model, which is related to the average loss of accuracy and the Gini index for the classifier model. The Gini index is a ‘measure of clutter’, represented as ‘MeanDecreaseGini’, meaning that the higher the measure, the higher the importance in the generated models, as values close to 0 for the Gini index imply more clutter and values close to 1 imply less clutter. The higher this measure, the more variability it contributes to the dependent variable.

Fourth, a similar analysis was performed in the subpopulation of patients with influenza and COVID-19 to obtain variables associated with COI in each subpopulation.

Statistical analysis was performed with R statistical software (v 4.4.1) R: The R Project for Statistical Computing (r-project.org).

## 3. Results

### 3.1. Whole Population

A total of 8902 patients admitted to the intensive care unit (ICU) were included in this study, of whom 3702 (41.6%) had influenza and 5200 (58.4%) had COVID-19. Table 1 presents the general characteristics of the patients. The cohort was predominantly male (66%), with a mean age of 60 years and an intermediate level of severity, as indicated by a mean APACHE II score of 14 points and a mean SOFA score of 5 points. The most prevalent comorbidities were obesity and chronic obstructive pulmonary disease (COPD), and there was a mean length of stay of 12 days and a crude ICU mortality of 25%.

A diagnosis of bacterial/fungal COI was made in 1263 patients (14.2%). The patients with COI were older and had higher levels of severity and inflammation than the patients without COI. In addition, except for obesity, comorbidities were more frequent in the COI patients, as well as the need for invasive mechanical ventilation (IMV), the presence of shock, and lenght of ICU stay. COVID-19 was associated with a lower frequency of COI. Conversely, patients with influenza were more frequent in the COI group. Finally, the crude mortality was significantly higher in patients with COI (33.4%) compared to those without this complication (24.5%, *p* < 0.001), as shown in Table 1.

*Streptococcus pneumoniae* (32.3%) was the most frequently isolated microorganism, followed by methicillin-sensitive *Staphylococcus aureus* (12.6%) and *Pseudomonas aeruginosa* (10.6%). Table 2 shows the number and proportion of the 1342 pathogens isolated in the 1263 critically ill patients with coinfection.

### 3.2. Factors Associated with COI in the Whole Population According to General Linear Model (GLM)

The associations between the presence of COI (dependent variable) and the independent variables were studied using multiple logistic regression. The independent variables that were statistically significant in the bivariate analysis were included in the GLM. To develop and validate the GLM, the population was randomly divided into two subsets: a development (Train) subset (70% of the total population, n = 6231) and a validation (Test) subset (30% of the total population, n = 2671).

The ROSE package was applied to address the imbalance between the groups, reducing the number of patients in the ‘Train’ sample from 6231 to 1734 (COI = 850 and No-COI = 884). This methodological option was only applied to the ‘Train’ subset.

The independent variables included in the model were: influenza, immunosuppression, pregnancy, hematological disease, chronic renal failure, chronic heart disease, COPD, presence of shock on ICU admission, lactate, PCT, C-reactive protein (C-RP), creatinine, lactate dehydrogenase (LDH), creatine phosphokinase (CPK), SOFA score, APACHE II score, and age.

Only influenza (OR = 2.01), shock on ICU admission (OR = 1.64), LDH (OR = 1.0), and age (OR = 1.01) were variables that were independently associated with the presence of COI (Figure 1). The Hosmer–Lemeshow test of the model resulted in a value of 0.48, suggesting that the model fit the data well. No collinearity was observed between the variables in the model.

### 3.3. GLM Validation

When applying the developed model to the Test subset, an acceptable model performance was evident, with an accuracy of 65%, a low sensitivity (23%) but a high specificity (91%), a positive predictive value (PPV) of 60%, and a negative predictive value of 67%. The precision was high (90%) but the area under the ROC curve was poor (AUC 0.68; 95% CI 0.65–0.71, *p* < 0.01) (Appendix A).

The cross-validation (K = 10) performed similarly to that for the original model, with an accuracy of 65%. More information on the cross-validation can be found in Appendix A. The study of the normality of the residuals showed that the distribution of the residuals was not normal (Appendix A). Finally, the RESET showed a *p*-value of 0.03, and, therefore, the hypothesis of the model’s linearity is not fulfilled.

### 3.4. Factors Associated with COI in the Whole Population According to No-Linear Model (Random Forest)

A random forest classifier (RFc) model was developed to study the contributions of confounding variables to the dependent variable (COI) in a non-linear way. The independent variables included in both RFc models were the same as in the GLM.

The RFc model had an OOB estimate of error rate of 36.9%. The occurrences of influenza, lactate, PCT and APACHE II were the four most important variables with respect to accuracy, while CPK, LDH, PCT, and C-RP were the four most important variables with respect to the Gini index (Figure 2). The accuracy of the RFc model was 65%.

### 3.5. Factors Associated with COI in the Influenza Cohort According to General Linear Model (GLM)

Of the 3702 influenza patients, 805 (21.7%) had a COI. The patient characteristics according to the presence or absence of COI are shown in Table 1. The patients with COI were older, had higher levels of severity and inflammation, and had higher levels of ICU mortality (30.3%) compared to those without COI (19.1%, *p* < 0.001).

For the development and validation of the GLM of COI risk factors in influenza patients, the population (n = 3072) was randomly divided into two sets, a training set with 70% of the population (n = 2592) and a test set with the remaining 30% (n = 1110). As a result of correcting the imbalance between the groups using the ROSE package, the number of patients in the ‘Train’ sample was reduced to 1108 (COI n = 564, No-COI n = 544).

The independent variables included in the model after the bivariate comparison between groups were immunosuppression, lactate, PCT, C-RP, SOFA score, APACHE II, age, sex, and creatinine. Only SOFA (OR = 1.05; 95%CI 1.0–1.08), APACHE II (OR = 1.03; 95%CI 1.0–1.05), PCT (OR = 1.01; 95% CI 1.0–1.02), and age (OR = 1.01; 95%CI 1.0–1.02) were variables associated with the presence of COI (Appendix A). The model performed poorly for prediction, with an accuracy of only 61%, a sensitivity of 31%, and a specificity of 86%. The AUC was also low at 0.66 (95%CI 0.62–0.70) (Appendix A). The examination of the residuals showed non-normality of the residuals, and non-linearity was confirmed by a RESET *p* < 0.001. Cross-validation (K = 10) of the model showed an even lower accuracy of 59%, with a sensitivity of 57% and a specificity of 64% (Appendix A).

### 3.6. Factors Associated with COI in the Influenza Cohort According to No-Linear Model (Random Forest)

The independent variables included in the RFc model were the same as in the GLM. The RFc model had an OOB estimate of error rate of 39.5%. Creatinine, APACHE II, PCT, and age were the four most important variables with respect to accuracy, while lactate, creatinine, C-RP, and PCT were the four most important variables with respect to the Gini index (Appendix A). The accuracy of the RFc model was 58%.

### 3.7. Factors Associated with COI in the COVID-19 Cohort According to General Linear Model (GLM)

Of the 5200 COVID-19 patients, 458 (8.8%) had a COI. The patient characteristics according to the presence or absence of COI are shown in Table 1. The patients with COI were older, had higher levels of severity and inflammation, and had higher levels of ICU mortality (38.9%) compared to those without COI (27.8%, *p* < 0.001).

To develop and validate the GLM of COI risk factors in COVID-19 patients, the population (n = 5200) was randomly divided into two sets, a training set with 70% of the population (n = 3640) and a test set with the remaining 30% (n = 1560). As a result of correcting the imbalance between the groups using the ROSE package, the number of patients in the training sample was reduced to 628 (COI n = 321, no-COI n = 307).

The independent variables included in the model after the bivariate comparison between groups were immunosuppression, lactate, shock, PCT, C-RP, CPK, SOFA score, APACHE II, age, COPD, and creatinine. Only shock (OR = 1.5; 95% CI 1.06–2.2) was associated with the presence of COI (Appendix A). The model performed poorly for prediction, with an accuracy of only 57%, a sensitivity of 11%, and a specificity of 92%. The AUC was also low at 0.58 (95% CI 0.54–0.63) (Appendix A). Examination of the residuals showed non-normality of the residuals and non-linearity was confirmed by a RESET *p* < 0.01. Cross-validation (K = 10) of the model showed an even lower accuracy of 50%, with a sensitivity of 48%, and a specificity of 70% (Appendix A).

### 3.8. Factors Associated with COI in the COVID-19 Cohort According to No-Linear Model (Random Forest)

The independent variables included in RFc model were the same as in the GLM. The RFc model had an OOB error rate estimate of 46.5%. CPK, shock, SOFA, APACHE II, and lactate were the top five variables in terms of accuracy, while CPK, C-RP, creatinine, PCT, and lactate were the top five variables in terms of the Gini index (Appendix A). The accuracy of the RFc model was 50.7%.

## 4. Discussion

Our results suggest that data obtained at the time of ICU admission cannot be used to adequately predict the presence of coinfection. Both the multiple linear and random forest models need to perform more adequately, suggesting that there are uncontrolled confounders that are not included in the models. However, the inflammatory variables (PCT, C-RP), severity (APACHEII, SOFA), and hemodynamic instability (shock, lactate, creatinine) appear to be important in most COI prediction models. Our study alerts physicians to the need to prioritize certain variables obtained on admission to the ICU in order to actively investigate the presence of coinfection and thus to try to optimize the administration of empirical antibiotic treatment.

While the prevalence of respiratory coinfection in viral infections varies according to different reports [5,6,11,12,13,14,19,21], influenza has a higher risk of COI than SARS-CoV-2 infection [5,6,7,8,11,13]. It is important to recognize viral infection in order to implement appropriate isolation and infection control measures and to facilitate the use of promising antiviral treatments. However, clinicians should not overlook the possibility of COI in patients with pandemic virus infection. It is difficult for clinicians to identify respiratory coinfections early because of their similar symptoms and signs to other types of infection, thus leading to a high rate of inappropriate prescriptions. This situation and the recommendations issued by scientific societies have led to a high overuse of antimicrobials, with the consequent appearance of microbial resistance, especially in Gram-negative bacilli [9,10,33].

The ability to predict the presence of respiratory COI based on the data available during ICU admission could facilitate the optimization of antimicrobial treatment. However, despite the development of predictive models related to patient mortality and the need for ICU admission or mechanical ventilation [34,35,36,37,38,39], there is a paucity of data from studies attempting to predict the presence of respiratory bacterial or fungal coinfection at the outset of ICU care.

A significant number of studies [34,35,36,37,38,39] have been conducted to predict the presence of coinfection in patients with COVID-19. However, these studies have several limitations. In a retrospective analysis of 235 patients with COVID-19, Su L et al. [34] found that the presence of invasive fungal infection (IFI) was associated with the use of broad-spectrum antibiotics (aOR 4.4), fever (aOR 2.3), log IL-6 concentration (aOR 1.2), and prone ventilation (aOR 2.3). The model’s performance was good, with an AUC of 81%, which is significantly better than our AUC of 68%. It should be noted, however, that there are several differences between the two studies. Firstly, it needs to be clarified whether an infection is a coinfection or a fungal superinfection. Secondly, the authors include *Candida* spp. in their definition of IFI, which was isolated in more than half of the cases (n = 33) that they studied. In conclusion, the reported performance is based on testing the model on the same population that was used to develop it. Therefore, it is likely that the model needs to be more balanced.

Recently, a retrospective multicenter study of 1977 patients with COVID-19 [35] reported that age (OR 1.02), male sex (OR 1.7), and APACHE IV (OR 1.01) at ICU admission were the variables associated with CAPA in their multivariate model. Although these factors were like those found in our study, unfortunately, the authors did not determine the performance of the generated model and, therefore, we cannot make a comparison or assess its true applicability.

Wang M. et al. [36] conducted a similar analysis in a population of 1778 patients with confirmed cases of SARS-CoV-2 infection, with a coinfection rate of 5%. The machine learning models (GLM and RF) that they developed to investigate risk factors associated with coinfection demonstrated robust predictive capabilities. Their algorithm showed that comorbidities (diabetes, neurological diseases), invasive procedures (central venous catheter [CVC], urinary catheter [UC]), baseline levels of inflammatory markers (IL-6, PCT), and creatinine were associated with an increased risk of bacterial/fungal coinfection, with an AUC of 87% for their GLM and 88% for their RF, higher values than those observed in our data. This discrepancy in performance outcomes may be attributed to the inclusion of outcome and treatment variables in the models proposed by Wang et al. [36], which differ from the variables considered at ICU admission, as seen in our models. Furthermore, our study is focused exclusively on respiratory COI. The presence of risk factors such as CVC or UC in the referenced research suggests including other types of coinfection.

A nationwide retrospective population-based study involving more than 200,000 hospitalized patients in Spain [37] who had an overall incidence of coinfection of 2% revealed that age, male sex, smoking, obesity, COPD, and metabolic disorders were the factors associated with coinfection in a multivariate analysis. Regrettably, the performance of the model is not presented, which hinders comparison of the developed model’s performance to that of ours.

The study by Delhommeau G et al. [13] compared incidences of pulmonary bacterial COI in patients with confirmed cases of both influenza and COVID-19 on admission to the ICU using a multicenter database. The authors observed that the prevalence of COI was considerably higher in patients with influenza (24.8%) compared to those with COVID-19 (8.4%). Furthermore, in the latter group, cirrhosis (OR 3.5) was identified as the sole independent risk factor for bacterial COI, suggesting that liver dysfunction significantly compromises the immune response, thereby increasing the susceptibility to bacterial infections. In contrast, immunosuppression (OR 0.34) and obesity (0.29) were identified as negative factors associated with the likelihood of COI in influenza patients. These findings highlight the complex and distinct immunopathological responses in influenza and COVID-19, underscoring the necessity for tailored approaches in managing coinfections in these conditions.

Immunosuppression, while generally increasing the risk of infections, might lead to less aggressive inflammatory responses, which could paradoxically reduce the likelihood of COI in certain viral infections.

Regrettably, Delhommeau G et al. [13] included nosocomial infections in their definition of respiratory COI and did not present data on their model’s performance, limiting the ability to generalize their findings. Future studies should differentiate between community-acquired and hospital-acquired infections to provide a clearer understanding of COI dynamics and develop more accurate predictive models. Additionally, exploring the temporal relationship between viral infections, such as influenza and COVID-19, and the onset of bacterial COI could offer insights into the optimal timing for prophylactic and therapeutic interventions.

Giannella M. et al. [39] developed a COI prediction score using logistic regression, considering only three variables with different cut-off points (PCT, WBC, and Charlson index). Their model performs adequately (AUC 83%) in classifying patients into low, intermediate, and high risk groups for COI. While the score is a valuable addition to the field, it should be noted that the authors include COI infections other than respiratory infections in their definition, with urinary tract infections accounting for almost half of the cases. This limits the applicability of the score in predicting respiratory COI.

To the best of our knowledge, there are no reports in the literature of predictive models for pulmonary COI in pandemic influenza patients. This is because the influenza A (H1N1)pdm09 pandemic occurred when machine learning techniques were not widely available.

Our group published a prospective observational study conducted between 2009 and 2015 in a large cohort of intensive care units in Spain [5]. In just over 2900 patients with influenza A (H1N1)pdm09, respiratory COI was diagnosed in 16.6% of patients. The likelihood of coinfection increased with age (aOR 1.01), previous HIV infection (aOR 2.6), and immunosuppressive medication (aOR 1.4), but unfortunately, we did not develop predictive models at that time.

Masia M et al. [40] showed, in more than 100 patients diagnosed with pandemic influenza A, that patients with pneumococcal COI were more likely to have confusion, a CURB-65 score > 1, and higher CRP levels (255 mg/L vs. 89 mg/L) than those without pneumococcal coinfection.

Recently, our group [29] published data on factors associated with respiratory *Aspergillus* spp. COI in patients with pandemic influenza. In a population of 3700 patients, regression modelling showed that male sex (OR: 2.81), asthma (OR: 4.56), nosocomial influenza infection (OR: 2.40), hematological malignancies (OR: 4.39), HIV (OR: 3.83), and other immunosuppression (such as chronic corticosteroid treatment or chemotherapy, OR: 4.87) were independently associated with *Aspergillus* spp. COI. In addition, using the CHAID (chi-squared automatic interaction detection) automated decision tree, hematological disease, with an incidence of 3.3%, was shown to be the variable that was most closely associated with *Aspergillus* spp. COI, followed by other immunosuppression as the second most crucial variable.

Experts have urged clinicians not to neglect the principles of antimicrobial stewardship during pandemics to avoid a worsening public health crisis related to antimicrobial resistance [9,10,11]. The challenge for clinicians is to differentiate patients with respiratory viral infections who may benefit from prompt antibiotic treatment from those with a low risk of COI in whom antibiotic pressure should be avoided. Our group [6,15,16,17] has highlighted the potential role of biomarkers such as C-reactive protein (CRP) and procalcitonin (PCT) in this discrimination process. However, all of these have been described as having low specificity and limited positive predictive value.

By using scientific statistical methods to estimate the risk of respiratory bacterial/fungal COI associated with influenza/COVID-19, we aim to assess the likelihood of an individual having a COI at the time of initial ICU care. These models could help in allowing doctors to intervene earlier, optimize antibiotic prescribing, or provide appropriate patient care. Therefore, the establishment of accurate predictive models is of practical importance for clinical work and is beneficial for identifying high-risk patients and their accurate prevention and follow-up. However, developing models using machine learning techniques requires reliable data and the inclusion of all confounding factors related to the dependent variable in order to demonstrate adequate performance. In addition, many situations in medicine have a non-linear association, so non-linear models such as random forest can better predict different events.

To our knowledge, our study is the first to attempt to develop a predictive model of respiratory COI in a large population of critically ill patients with respiratory failure secondary to pandemic viral infection. Despite applying machine learning techniques, the developed models did not perform adequately in the validation process. However, the development of the models has allowed the identification of variables related to the severity or inflammatory states of patients that have a strong association with the presence of COI. With the clinical and laboratory data available at the time of ICU admission, the prediction of coinfection cannot be higher than 70%.

Our study has several strengths. It is a multicenter study with many patients, all with a microbiological diagnosis of coinfection, and a robust statistical analysis using machine learning techniques. However, it also has several limitations that must be acknowledged. Firstly, it is a retrospective national study, merging two large databases collected prospectively over more than 15 years. During this time, organ support techniques have evolved and changed significantly, which may impact prognoses. However, the aim of this study is not to evaluate the evolution of patients but to try to develop a predictive model of coinfection.

Second, this is a secondary analysis, so the possibility of bias related to uncontrolled confounding variables cannot be excluded. In addition, the inclusion of outcome variables (e.g., type of treatment, ventilatory support modalities, etc.) could improve the performance of the models developed herein. However, this study aimed to provide a method for the early prediction of the risk of respiratory coinfection.

Third, we chose the ROSE package with the under-sampling option because we felt it was the best option to reduce the risk of bias towards the majority class. However, the loss of potentially important data due to the removal of data from the majority class is a risk that we cannot rule out and could affect the performance of the models.

Fourth, it cannot be excluded that some patients received antimicrobial treatment prior to ICU admission (data not available). However, as the median time to ICU admission was only 1 day, the effect of this short treatment on respiratory specimen cultures should be small. Nevertheless, the possibility of a higher incidence of coinfection cannot be excluded.

Fifth, we did not explicitly set hyperparameters for the machine learning model that could improve the performance of the model. In future work, we plan to incorporate a more systematic search, such as a grid search or random search with cross-validation, while being mindful of the risk of overfitting, to ensure that we achieve the best possible balance between model accuracy and generalizability.

Finally, although this is a huge and homogeneous cohort of critically ill patients, it is predominantly a national sample, so these results can only be transferred to other countries or regions with prior validation.

## 5. Conclusions

Developing early prediction models for respiratory coinfection in patients with pandemic viral infections using machine learning techniques has resulted in a prediction accuracy of only 70%. However, the models developed herein have identified factors such as age, PCT, C-RP, CPK, APACHE II, SOFA, and shock which are strongly associated with coinfection. Therefore, the presence of these risk factors in a patient with a confirmed viral infection obliges the intensivist to actively exclude the presence of this complication. To enhance the predictive capability of the models developed herein, future prediction studies should incorporate a greater number of confounding variables.

## Figures and Tables

**Figure 1 antibiotics-13-00968-f001:**
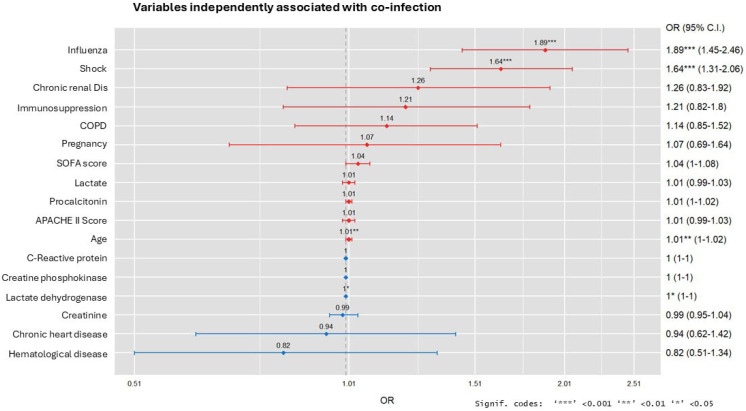
Odds ratio (OR) plot of variables associated with the presence of coinfection on admission to the ICU.

**Figure 2 antibiotics-13-00968-f002:**
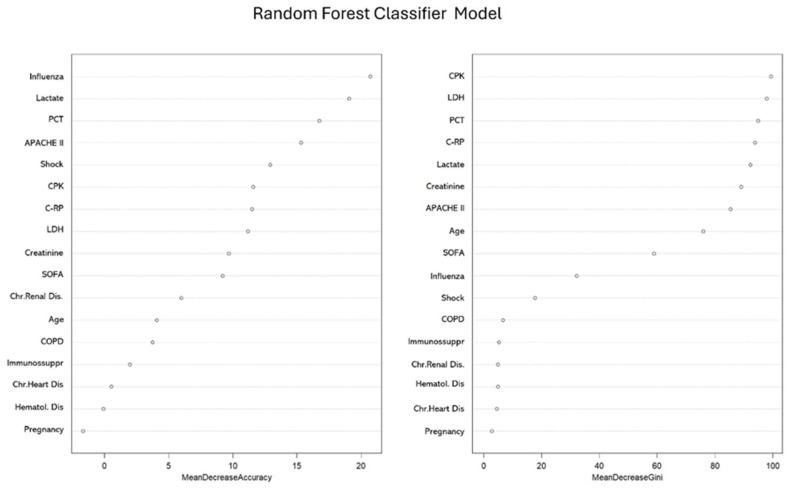
Importance of variables according to the random forest classifier model. (PCT: procalcitonin; CPK: creatine phosphokinase; C-RP: C-reactive protein; LDH: lactate dehydrogenase; COPD: chronic obstructive pulmonary disease; Immunossuppr: immunosuppression; Chr.: chronic; Dis: disease).

**Table 1 antibiotics-13-00968-t001:** Baseline characteristics of the 8902 patients included in the analysis, categorized by the presence of community-acquired coinfection (COI) in the whole population and according to influenza or COVID-19 virus infection at ICU admission.

Variables #	Whole Population (n = 8902)			Influenza Cohort (n = 3702)			COVID-19 Cohort (n = 5200)		
	Total	No-COI (n = 7639)	COI (n= 1263)	Total	No-COI (n = 2897)	COI (n = 805)	Total	No-COI (n = 4742)	COI (n = 458)
General characteristics									
Age, years	60 (49–70)	60 (49–69)	62 (51–72) ***	55 (43–67)	54 (42–66)	59 (47–72) ***	63 (54–71)	63 (54–71)	65 (57–72) ***
Male sex	5855 (66.1)	5012 (65–6)	843 (66.7)	2203 (59.5)	1698 (58.6)	505 (62.7) *	3652 (70.2)	3314 (69.9)	338 (73.8)
APACHE II score	14 (10–19)	14 (10–18)	17 (12–22) ***	15 (11–21)	15 (11–20)	19 (14–24) ***	13 (10–17)	13 (10–17)	14 (11–18) ***
SOFA score	5 (3–7)	5 (3–7)	6 (4–9)	5 (4–8)	5 (4–8)	7 (4–10) ***	4 (3–7)	4 (3–7)	5 (4–8) ***
Gap-ICU, days	1 (1–3)	1 (1–3)	1 (0–2) ***	3 (1–6)	1.0 (1.0–2.0)	1.0 (1.0–2.0)	2 (0–4)	2 (0–4)	1 (0–3) *
COVID	5200	4742 (61.1)	458 (36.3) ***	-----	-----	-----	-----	-----	-----
Influenza	3702	2897 (37.9)	805 (63.7) ***	-----	-----	-----	-----	-----	-----
Laboratory									
WBC ×10^3^	8.6 (5.7–12.5)	8.6 (5.8–12.3)	8.9 (5.8–13.7)	8.1 (4.6–12.4)	8.0 (4.7–12.0)	8.8 (3.6–14.3)	8.9 (6.4–12.5)	8.9 (6.4–12.4)	9.0 (6.1–13.1)
LDH U/L	542 (403–687)	537 (401–686)	566 (415–697) *	599 (457–750)	600 (456–754)	594 (449–737)	501 (380–632)	500 (379–632)	514 (390–631)
C-RP mg/mL	19.6 (9.8–34.7)	18.3 (9.3–32.8)	27.0 (14.3–64.2) ***	36 (19–84)	34 (17–20)	42 (23–100) ***	14 (7–22)	14 (7.2–22.5)	14 (8.6–24.6) ***
PCT ng/mL	0.8 (0.20–5.67)	0.73 (0.19–4.14)	3.4 (0.56–19.3) ***	7.2 (1.9–22.0)	6.2 (1.7–20.7)	12.2 (3.2–26.4) ***	0.26 (0.11–0.72)	0.25 (0.11–0.70)	0.30 (0.14–0.8) ***
Creatinine mg/dL	0.89 (0.70–1.23)	0.87 (0.69–1.17)	1.0 (0.75–1.57) ***	1.0 (0.70–1.46)	0.9 (0.7–1.4)	1.2 (0.8–1.9) ***	0.8 (0.7–1.0)	0.8 (0.6–1.0)	0.8 (0.7–1.1) ***
CPK	216 (100–420)	208 (98–410)	272 (119–490) ***	331 (145–585)	327 (146–578)	344 (138–629)	170 (83–321)	169 (83–320)	193 (95–335) *
Lactate mmol/L	2.0 (1.4–3.3)	1.9 (1.3–3.1)	2.8 (1.8–4.2) ***	3.2 (2.8–4.7)	3.2 (2.3–4.7)	3.6 (2.7–5.0) ***	1.5 (1.1–2.1)	1.5 (1.1–2.0)	1.6 (1.2–2.2) ***
Comobidities									
COPD	1281 (14.3)	1022 (13.4)	259 (20.5) ***	908 (24.5)	696 (24.0)	212 (26.3)	373 (7.2)	326 (6.8)	47 (10.3) *
Asthma	698 (7.8)	595 (7.8)	103 (8.1)	367 (10.0)	296 (10.2)	71 (8.8)	331 (6.4)	299 (6.3)	32 (6.9)
Chr. Heart Dis	623 (7.0)	501 (6.6)	122 (9.6) ***	447 (12.0)	347 (12.0)	100 (12.4)	176 (3.4)	154 (3.2)	22 (4.8)
Chr.Renal Dis.	595 (6.7)	486 (6.4)	109 (8.6) ***	314 (8.5)	235 (8.1)	79 (9.8)	281 (5.4)	251 (5.3)	30 (6.5)
Hematologic Dis.	436 (4.9)	343 (4.5)	93 (7.4) ***	272 (7.3)	202 (6.7)	70 (8.7)	164 (3.1)	141 (2.9)	23 (5.0) *
Pregnancy	480 (5.4)	364 (4.7)	116 (9.2) ***	460 (12.4)	344 (11.9)	116 (14.4)	20 (0.38)	20 (0.4)	0 (0.0)
Obesity	3046 (34.2)	2677 (35.0)	369 (29.2) ***	1178 (31.8)	985 (34.0)	193 (24.0) ***	1868 (35.9)	1692 (35.7)	176 (38.4)
IS	711 (7.9)	564 (7.4)	147 (11.6) ***	419 (11.3)	305 (10.5)	114 (14.2) **	292 (5.6)	1 (0.02)	1 (0.22)
Treatments and procedures									
EAT	7410 (83.2)	6228 (81.5)	1182 (93.6) ***	3240 (87.5)	2452 (84.6)	788 (97.9) ***	4170 (80.2)	3776 (79.6)	394 (86.9) ***
Corticosteriods	5275 (59.3)	4530 (59.3)	745 (59.0)	1438 (38.8)	1048 (36.2)	390 (48.4) ***	3837 (73.8)	3482 (73.4)	355 (77.5)
IMV	5998 (67.4)	3512 (46.0)	740 (58.6) ***	2072 (56.0)	1566 (54.1)	506 (62.9) ***	3926 (75.5)	3510 (74.0)	416 (90.8)
AKI	1435 (16.1)	1081 (14.2)	354 (28.0) ***	904 (24.4)	608 (21.0)	296 (36.8) ***	531 (10.2)	473 (9.9)	58 (12.7)
Prone IMV	4064 (45.6)	3469 (45.4)	595 (47.1)	1101 (29.7)	837 (28.9)	264 (32.8) *	2963 (57.0)	2632 (55.5)	331 (72.3) ***
Shock	3549 (39.9)	2827 (37.0)	722 (57.2) ***	1899 (51.3)	1363 (47.0)	536 (66.6) ***	1650 (31.7)	1464 (30.9)	186 (40.6) ***
Outcomes									
LOS ICU, days	13 (6–23)	12 (6–23)	14 (6–27) ***	10 (4–18)	10.0 (4–18)	10.0 (5–19)	15 (8–27)	14 (7–26)	23 (13–37) ***
IMV days	12 (6–23)	12 (6–23)	13 (7–25)	8 (3–17)	8 (3–16)	10 (4–18) ***	15 (8–27)	15 (8–27)	19 (11–33) ***
ICU mortality	2294 (25.8)	1872 (24.5)	422 (33.4) ***	796 (21.5)	552 (19.1)	244 (30.3) ***	1498 (28.8)	1320 (27.8)	178 (38.9) ***

# Continuous variables are shown as median values and percentiles Q1–Q3. Categorical variables are shown as number of cases (n) and percentage (%). (LDH: lactate dehydrogenase, C-RP: C-reactive protein, CPK: creatine phosphokinase, PCT: procalcitonin, IS: immunosuppression, EAT: empirical antibiotic treatment; IMV: invasive mechanical ventilation, AKI: acute kidney injury, LOS length of stay, ICU: intensive care units). Signification: * *p* < 0.5; ** *p* < 0.01; *** *p* < 0.001; *p*-value was used for the comparison between no coinfection (reference) and coinfection (COI) in the whole population and in each cohort.

**Table 2 antibiotics-13-00968-t002:** Number and proportion of 1342 pathogens isolated from 1263 critically ill patients with coinfection.

Microorganism	COI Whole Population n = 1263	Influenza Cohort n = 805	COVID-19 Cohort n = 458
*Streptococcus pneumoniae*, n (%)	433 (32.3)	367 (44.8)	66 (12.4)
*Staphylococcus aures* Methicillin-sensitive, n (%)	172 (12.8)	99 (12.1)	73 (13.8)
*Pseudomonas aeruginosa*, n (%)	143 (10.6)	56 (6.9)	87 (16.4)
*Aspergillus* spp., n (%)	78 (5.8)	42 (5.2)	36 (6.8)
*Escherichia coli*, n (%)	69 (5.1)	23 (2.8)	46 (8.7)
*Klebsiella* spp., n (%)	66 (4.8)	19 (2.3)	47 (8.8)
*Haemophilus influenzae*, n (%)	61 (4.5)	38 (4.7)	23 (4.3)
*Staphylococcus aures* Methicillin-resistant, n (%)	56 (4.2)	34 (4.1)	22 (4.1)
*Streptococcus pyogenes*, n (%)	45 (3.3)	45 (5.6)	0 (0.0)
*Enterobacter* spp., n (%)	30 (2.3)	4 (0.5)	26 (4.9)
*Serratia* spp., n (%)	23 (1.6)	8 (1.0)	15 (2.8)
*Staphylococcus hominis*	22 (1.6)	6 (0.7)	16 (3.0)
*Stenotrophomonas maltophilia*, n (%)	21 (1.5)	6 (0.7)	15 (2.8)
*Moraxella catarrhalis*, n (%)	15 (1.1)	12 (1.4)	3 (0.6)
*Acinetobacter baumannii*, n (%)	14 (1.0)	14 (1.7)	0 (0.0)
*Chlamydia* spp., n (%)	10 (0.7)	5 (0.6)	5 (0.9)
*Mycoplasma* spp., n (%)	10 (0.7)	5 (0.6)	5 (0.9)
*Staphylococcus haemolyticus*, n (%)	10 (0.7)	0 (0.0)	10 (1.8)
*Streptococcus agalactiae*, n (%)	5 (0.4)	0 (0.0)	5 (0.9)
*Coxiella burnetii*, n (%)	5 (0.4)	0 (0.0)	5 (0.9)
*Pneumocystis jirovecii*, n (%)	5 (0.4)	5 (0.6)	0 (0.0)
*Morganella morganii*, n (%)	4 (0.3)	2 (0.2)	2 (0.4)
*Proteus* spp., n (%)	4 (0.3)	0 (0.0)	4 (0.8)
*Corynebacterium* spp., n (%)	4 (0.3)	0 (0.0)	4 (0.8)
*Citrobacter* spp., n (%)	4 (0.3)	0 (0.0)	4 (0.8)
Others, n (%)	40(3.0)	29 (3.5)	11 (2.0)
Total, n (%)	1342(100)	819 (100)	530 (100)

Note: Of the total number of patients with isolation, 1263 (94.1%) presented isolation of 1 micro-organism, 70 (5.2%) of 2 micro-organisms, and 9 (0.7%) of three micro-organisms.

## Data Availability

The corresponding author (AR) had full access to all the data in the study and takes responsibility for the integrity of the data and the accuracy of the data analysis. All authors approved the final version of the manuscript. The views expressed in this article are those of the authors and not necessarily those of the SEMICYUC. The data supporting the conclusions of this study are available from the Spanish Society of Critical Care (SEMICYUC), but restrictions apply to the availability of these data, which were used under SEMICYUC authorization for the present study and are therefore not publicly available. However, the data can be obtained from the corresponding author (AR) upon reasonable request and with the permission of SEMICYUC.

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
