# Peer review of "A Machine Learning Approach to Determine Risk Factors for Respiratory Bacterial/Fungal Coinfection in Critically Ill Patients with Influenza and SARS-CoV-2 Infection: A Spanish Perspective"

_antibiotics, 2024, doi:10.3390/antibiotics13100968_

Round 1
Reviewer 1 Report
Comments and Suggestions for Authors
The manuscript by Alejandro Rodríguez et al. aimed to explore the use of machine learning models to identify risk factors for bacterial and fungal coinfections in critically ill patients with influenza and SARS-CoV-2. This study is on a unique cohort of patients in Spain, it used the methods of logistic regression and random forest models to predict coinfections at ICU admission. The topic of pandemic preparedness and antimicrobial stewardship is highly relevant. However, the manuscript could be improved to better clarity and impact.
Specific comments
1. Include specific data points or results when discussing the models. Instead of stating that "performance was modest," you can directly mention the AUC values and compare them with other studies for context.
2. The tables are generally informative, but their readability can be improved by ensuring consistent formatting and clearer labeling.
3. The discussion section could further explore the broader implications of the findings, particularly regarding clinical practice and future research.
Author Response
Dear Reviewer, thank you very much for your comments and contributions to improve the manuscript. The following is a point-by-point response to the questions.
Comments and Suggestions for Authors
The manuscript by Alejandro Rodríguez et al. aimed to explore the use of machine learning models to identify risk factors for bacterial and fungal coinfections in critically ill patients with influenza and SARS-CoV-2. This study is on a unique cohort of patients in Spain, it used the methods of logistic regression and random forest models to predict coinfections at ICU admission. The topic of pandemic preparedness and antimicrobial stewardship is highly relevant. However, the manuscript could be improved to better clarity and impact.
Specific comments
- Include specific data points or results when discussing the models. Instead of stating that "performance was modest," you can directly mention the AUC values and compare them with other studies for context.
REPLY: Thank you for your comment. Unfortunately, we do not understand what the reviewer means by directly mentioning AUC values. In the validation of the models, the AUC is clearly specified in the text and more data on the performance of each model is added in the supplement.
- The tables are generally informative, but their readability can be improved by ensuring consistent formatting and clearer labeling.
REPLY: Thank you for your comment. We agree with the reviewer that the current format of the table is somewhat confusing. But this current word format submitted for review will be edited appropriately in the final manuscript and we believe that this situation will be remedied.
- The discussion section could further explore the broader implications of the findings, particularly regarding clinical practice and future research.
REPLY: Thank you for your important comment. We have reinforced the message about the clinical impact of our results in the first paragraph of the discussion “Our study alerts physicians to the need to prioritize certain variables obtained on admission to the ICU in order to actively investigate the presence of co-infection and thus try to optimize the administration of empirical antibiotic treatment”. In addition, we have highlighted the need to include a greater number of confounding variables in future studies in the final concussion “To enhance the predictive capability of the models developed, future prediction studies should incorporate a greater number of confounding variables.”
Reviewer 2 Report
Comments and Suggestions for Authors
The work presented by Rodríguez et al is well written, the science is sound, and the results are relevant to the field. It is an interesting approach that uses a large sample, and adequate statistics, which makes it robust. Although the proposed model only resulted in a 70% correct prediction, several risk factors have been identified in the present study, which might help in the treatment of patients in the ICU with confirmed viral respiratory infections.
Author Response
Comments and Suggestions for Authors
The work presented by Rodríguez et al is well written, the science is sound, and the results are relevant to the field. It is an interesting approach that uses a large sample, and adequate statistics, which makes it robust. Although the proposed model only resulted in a 70% correct prediction, several risk factors have been identified in the present study, which might help in the treatment of patients in the ICU with confirmed viral respiratory infections.
REPLY: Thank you very much for your positive assessment of our studio
Reviewer 3 Report
Comments and Suggestions for Authors
Novelty
This aim of the study is to develop different prediction models using machine learning to predict the presence of COI at ICU admission based on an extensive national database. This paper may possibly contribute to understanding the factors associated with co-infection, which may help to assist clinicians in ruling out this complication upon admission to the ICU.
Scope
The work focuses on identifying risk factors associated with co-infection, hence it may be related to journal scope.
Significance
The results are interpreted appropriately, and the conclusions are supported by the results, but there are concerns on the reliability of the results as the statistical methods are not properly justified.
Quality
The article is written in an appropriate way, but the authors should consider adding the following:
· References at lines 111-113 to support your definitions of community-acquired COI
· References at lines 151-152 and clarify which variables have been identified as important from a clinical perspective, which were included in the final model.
· Footnote for Table 2: Number and proportion of 1,342 pathogens isolated from 1,263 critically ill patients with co-infection to explain these patients may have one or more pathogen isolated.
· Include definitions of abbreviations and asterisks in Figure 1.
· Include definitions of abbreviations in Figure 2. Figure 2 can be improved with higher resolution.
· Include limitations of the imbalanced nature of the datasets.
Scientific Soundness
The study appears to be designed in line with the research objectives. However, there are concerns regarding the selection of statistical methods, which may require further justifications:
· Please provide a justification for the selection of the ROSE (Random Over-Sampling Examples) package to address the imbalance between the groups, instead of using other methods such as SMOTE or a combination of different packages.
· Please provide a justification for the selection of the Random Forest classifier model (RFc). Additionally, please provide a justification for the selection of 500 random trees, with a minimum number of 15 variable per tree. Please provide the cross-validation results.
· Please provide justification for selecting a 70% training set and a 30% test set.
Interest to the Readers
This manuscript may be of relevance to the readers, to help clinicians in understanding risk factors associated with co-infection.
Overall Merit
The work can potentially advance current knowledge in identifying risk factors associated with co-infection. However, there are concerns regarding the statistical methods used, which may affect the interpretations of results and conclusions drawn.
Comments on the Quality of English Language
Academic writing can be improved.
Author Response
Dear Reviewer, thank you very much for your comments and contributions to improve the manuscript. The following is a point-by-point response to the questions.
Quality
The article is written in an appropriate way, but the authors should consider adding the following:
- References at lines 111-113 to support your definitions of community-acquired COI
REPLY: Thank you for your comment. The definition of co-infection used was the one published by our group in the referenced studies (ref 5,6,7 and 17) . These publications support the definition used in this manuscript.
- References at lines 151-152 and clarify which variables have been identified as important from a clinical perspective, which were included in the final model.
REPLY: Thank you for your comments. The variables incorporated were at the discretion of the researchers and not related to the literature. Only the SOFA score was a variable considered of interest to include in the model despite not achieving significance in the comparative table (Table 1). We have modified the sentence for a better understanding following the reviewer's indications. The sentence now reads: “Furthermore, the SOFA score was incorporated into the final model, deemed a crucial factor from a clinical standpoint despite failing to achieve statistical significance in the comparative table”
- Footnote for Table 2: Number and proportion of 1,342 pathogens isolated from 1,263 critically ill patients with co-infection to explain these patients may have one or more pathogen isolated.
REPLY: Thank you for your comment. We have added a footnote to the table with the identification percentages of 1 or more microroganisms following the reviewer's indications.”Of the total number of patients with isolation 1263 (94.1%) presented isolation of 1 micro-organism, 70 (5.2%) of 2 micro-organisms and 9 (0.7%) of three micro-organisms”
- Include definitions of abbreviations and asterisks in Figure 1.
REPLY: Thank you for your comment. The significance codes have been added to figure 1.
- Include definitions of abbreviations in Figure 2. Figure 2 can be improved with higher resolution.
REPLY: Thank you for your comment and apologies for the inadvertent error. We have added the definition of abbreviations in the footer of Table 2 as suggested by the reviewer.
- Include limitations of the imbalanced nature of the datasets.
REPLY: Thank you for your comment. We have added a paragraph on limitations following the reviewer's indications. “Third, we chose the ROSE package with the undersampling option because we felt it was the best option to reduce the risk of bias towards the majority class. However, the loss of potentially important data due to the removal of data from the majority class is a risk that we cannot rule out and could affect the performance of the models.”
Scientific Soundness
The study appears to be designed in line with the research objectives. However, there are concerns regarding the selection of statistical methods, which may require further justifications:
- Please provide a justification for the selection of the ROSE (Random Over-Sampling Examples) package to address the imbalance between the groups, instead of using other methods such as SMOTE or a combination of different packages.
REPLY: Thanks for your very good comment. The reviewer is right to point out that there are other ways of dealing with unbalanced data. We chose to use the ROSE package with the 'under' option because we felt it was better to subsample or downsample to address the class imbalance in our study. We did not use SMOTE (Synthetic Minority Oversampling Technique) because it is an oversampling technique that creates artificial data through interpolation. Furthermore, SMOTE is only suitable for continuous variables, although there are extensions of SMOTE that deal with datasets with numeric and categorical variables, or with categorical variables only.
Although both undersampling and oversampling have advantages and limitations, we believe that the use of SMOTE can lead to overfitting due to duplicate sampling of minority classes and can produce synthetic data samples that are unrealistic or unrepresentative of the true distributions.
On the other hand, subsampling is an appropriate option when dealing with large data sets. By using all instances of the rare class and randomly removing instances of the majority class, the dataset can be transformed into a balanced dataset with equal representation of both classes. This can help solve the problem of unbalanced data and improve the performance of the model without overfitting it.
- Please provide a justification for the selection of the Random Forest classifier model (RFc). Additionally, please provide a justification for the selection of 500 random trees, with a minimum number of 15 variable per tree.
REPLY: The Random Forest classifier model (RFc) was chosen for its ability to handle large datasets with high dimensionality, as it is an ensemble learning technique. By aggregating the predictions of multiple decision trees, Random Forest can reduce overfitting and improve generalisation performance. This is particularly useful in situations where the relationship between features and the target variable is complex and non-linear, as the Random Forest model can capture these intricate relationships.
The selection of 500 random trees, with a minimum of 15 variables per tree, was chosen to balance accuracy and efficiency. Although the original study suggested 100 trees (1), today, with advances in the speed of analysis, a larger number of trees in the forest (500) allows the model to capture different patterns and reduces the risk of overfitting. In addition, setting a minimum number of variables per tree that is not too small helps to introduce randomness and prevent individual trees from dominating the final prediction.
Overall, the combination of a random forest classifier with 500 random trees and a minimum of 15 variables per tree provides a robust and reliable model for handling complex, high-dimensional datasets.
- LEO BREIMAN. Random Forests. Machine Learning, 45, 5–32, 2001
. Please provide the cross-validation results.
REPLY: Thank you for your comment. The results of the cross-validation are shown in the supplementary material.
- Please provide justification for selecting a 70% training set and a 30% test set.
REPLY: Thank you for your comment. We decided on this split based on the usual 80%/20% or 70%/30% recommendations for internal validation. Typically, for very large databases, it is suggested that 80% of the dataset be used to train the model and the remaining 20% be used for evaluation, and in some cases 70% of the dataset for training and 30% for validation when the dataset is smaller. We found that the latter partitioning was the most appropriate for our database.
Interest to the Readers
This manuscript may be of relevance to the readers, to help clinicians in understanding risk factors associated with co-infection.
Overall Merit
The work can potentially advance current knowledge in identifying risk factors associated with co-infection. However, there are concerns regarding the statistical methods used, which may affect the interpretations of results and conclusions drawn.
REPLY: Thank you for your comment. We hope that we have been able to clarify the doubts related to the methodology and the statistical analysis carried out.
Comments on the Quality of English Language
Academic writing can be improved.
REPLY: Thank you for your comment. The language was optimised
Reviewer 4 Report
Comments and Suggestions for Authors
The use of machine learning used in the study confirms numerous reports that indicate the same risk factors described in previously published studies.
The thesis presented in the Introduction lines 57-61 is very radical and even accusatory. How can a doctor avoid this error based on your study?
On what basis was the selection of risk factors presented in the study made?
No definition of co-infection. No mention of suprainfection.
Most patients are admitted to the ICU after a few days of treatment, with empirical antibiotic therapy initiated. Could this have influenced the presented results?
Why is the abbreviation BFC used in the Abstract, while in the text COI?
Author Response
Dear Reviewer, thank you very much for your comments and contributions to improve the manuscript. The following is a point-by-point response to the questions.
The use of machine learning used in the study confirms numerous reports that indicate the same risk factors described in previously published studies.
1.- The thesis presented in the Introduction lines 57-61 is very radical and even accusatory. How can a doctor avoid this error based on your study?
REPLY: Thank you for your important comment. It was not our intention to make an accusation, but only to describe a situation observed during the pandemic and published by our group. We believe that it is because of this bad experience that we decided to carry out the present study to try to improve or optimise the use of antimicrobials in patients with viral infections. The contribution of our study to physicians is to alert them to the recognition of certain clinical factors present on admission to the ICU that are more strongly associated with the possibility of co-infection.
2.- On what basis was the selection of risk factors presented in the study made?
REPLY: Thank you for your comment: As indicated in the Materials and Methods subtitle Variables “ All variables with statistical significance (p<0.05) in the bivariate comparison between groups were included in the GLM (generalised linear model). In addition, the SOFA score, which was considered a clinically important factor although it did not reach statistical significance in the comparison table, was included in the final model”.
3.- No definition of co-infection. No mention of suprainfection.
REPLY: Thank you for your comment. The definition of co-infection is presented in material and method under the subheading definitions: “Coinfection was suspected if a patient presented with signs and symptoms of lower respiratory tract infection, with radiographic evidence of a pulmonary infiltrate with no other known cause (5-7).
Coinfection had to be confirmed by laboratory testing using Centers for Disease Control and Prevention (CDC) criteria (5,6,17) Only respiratory infection microbiolog-ically confirmed with a respiratory specimen or serology obtained within two days of ICU admission was considered community-acquired COI.
Regarding superinfection, we have called it nosocomial infection as it appears in definitions. “Lower respiratory tract infections (LTRI) diagnosed after two days of ICU admission were considered nosocomial and excluded from the present study.”
However, following the reviewer's indications, we have added the term superinfection in the sentence. Now the sentence reads “Lower respiratory tract infections (LTRI) diagnosed after two days of ICU admission were considered nosocomial superinfection and excluded from the present study”
Most patients are admitted to the ICU after a few days of treatment, with empirical antibiotic therapy initiated. Could this have influenced the presented results?
REPLY: Thank you for your interesting comment. It is true that some patients may have received antimicrobial treatment prior to ICU admission and this may decrease the possibility of bacterial isolation in respiratory samples. However, the median number of days to ICU has been only 1, so the impact of antimicrobial pre-treatment on cultures with such a short time of antibiotic treatment seems to be poor. However, we have added a paragraph on this aspect in limitations.
Why is the abbreviation BFC used in the Abstract, while in the text COI?
REPLY: Thank you for your comment. This error has been fixed
Round 2
Reviewer 3 Report
Comments and Suggestions for Authors
Quality
The quality has improved after revision. However, Figure 2 is still provided in low resolution.
Scientific Soundness
The study appears to be designed in line with the research objectives. However, there are still concerns regarding the selection of statistical methods, which may require further justifications:
- Can you please clarify further if you have tuned your optimal hyperparameter using your training set. If you have tuned your hyperparameter, please provide search range and relevant cross validation results for different combinations of hyperparameter. You have just randomly selected the values for each hyperparameter. How would you ensure that you have built the most accurate model.
- Can you please clarify further if you have tuned your data partition. If yes, please provide relevant cross validation results for different combinations of data partition. If not, please clarify further.
Interest to the Readers
This manuscript may be of relevance to the readers, to help clinicians in understanding risk factors associated with co-infection.
Overall Merit
There are concerns still regarding the statistical methods used, which may affect the interpretations of results and conclusions drawn.
Comments on the Quality of English Language
Improved
Author Response
REPLY: Thank you for your valuable comments. In this study, we did not perform an explicit hyperparameter setting for the machine learning model. However, it is important to clarify that the parameters were not randomly selected either. They were carefully chosen based on the comparative table of differences between the study groups and on best practices for creating robust models while avoiding over-fitting our data, in favour of generalisation. Our aim was to maintain a balance between model complexity and generalisability to unknown data.
We acknowledge the importance of hyperparameter optimization in improving model performance. However, it should also be noted that extensive hyperparameter tuning does not always guarantee better performance on unseen data, as it may lead to overfitting to the training set. In future work, we plan to incorporate a more systematic search, such as grid search or random search with cross-validation, while being mindful of the risk of overfitting, to ensure that we achieve the best possible balance between model accuracy and generalizability.
Finally, as reported by Hansika Hewamalage et al. (Forecast evaluation for data scientists: common pitfalls and best practices. Data Mining and Knowledge Discovery (2023) 37:788-832) if a model is poor and does not produce good forecasts, performing a validation to select hyperparameters, using any of the hyperparametrisation strategies, will be of limited value.
However, we agree with the reviewer that it can be seen as a limitation of the study, so have added a paragraph on this aspect in limitations.
Round 3
Reviewer 3 Report
Comments and Suggestions for Authors
Could you kindly revise the study limitation to the following:
Fifth, we did not explicitly set hyperparameters for the machine learning model that could improve the performance of the model. In future work, we plan to incorporate a more systematic search, such as grid search or random search with cross-validation, while being mindful of the risk of overfitting, to ensure that we achieve the best possible balance between model accuracy and generalizability.
Author Response
REPLY : Thank you for your comment. We have amended the judgment in accordance with your instructions.